# The Robust Multi-Scale Deep-SVDD Model for Anomaly Online Detection of Rolling Bearings

**DOI:** 10.3390/s22155681

**Published:** 2022-07-29

**Authors:** Linlin Kou, Jiaxian Chen, Yong Qin, Wentao Mao

**Affiliations:** 1State Key Laboratory of Rail Traffic Control and Safety, Beijing Jiaotong University, Beijing 100044, China; koulinlin5829@bjsubway.com (L.K.); koull01@163.com (Y.Q.); 2Beijing Mass Transit Railway Operation Corp. Ltd., Beijing 100044, China; 3College of Computer and Information Engineering, Henan Normal University, Xinxiang 453007, China; 12120823@bjtu.edu.cn

**Keywords:** incipient fault detection, robustness, reinforcement learning, anomaly detection

## Abstract

Aiming at the online detection problem of rolling bearings, the limited amount of target bearing data leads to insufficient model in training and feature representation. It is difficult for the online detection model to construct an accurate decision boundary. To solve the problem, a multi-scale robust anomaly detection method based on data enhancement technology is proposed in this paper. Firstly, the training data are transformed into multiple subspaces through the data enhancement technology. Then, a prototype clustering method is introduced to enhance the robustness of features representation under the framework of the robust deep auto-encoding algorithm. Finally, the robust multi-scale Deep-SVDD hyper sphere model is constructed to achieve online detection of abnormal state data. Experiments are conducted on the IEEE PHM Challenge 2012 bearing data set and XJTU-TU data set. The proposed method shows much greater susceptibility to incipient faults, and it has fewer false alarms. The robust multi-scale Deep-SVDD hyper sphere model significantly improves the performance of incipient fault detection for rolling bearings.

## 1. Introduction

As a kind of special mechanical parts, rolling bearing has a decisive impact on the operation and reliability of mechanical equipment [1]. Once damaged, it will cause major losses to industrial production and personal property. Detecting abnormalities in the early stages of bearing fault, and performing accurate and reliable detection and diagnosis, will help to take timely measures for maintenance and avoid major accidents. Incipient fault detection is a key link in the Prognostics and Health Management (PHM) for rolling bearings [2]. 

For signal analysis-based incipient fault detection methods, noise elimination and noise utilization are conducted at first for vibration signal. Then, time domain, frequency domain or time-frequency domain analysis are performed to extract and compare fault characteristics [3,4,5]. Vibration spectra were conducted in [6] with consideration of a set of different recurrence indicators to describe the response of the bearing to the optimal clearance. Acoustic emission and lubricating oil characteristics are also very helpful for condition monitoring of bearings. Liu et al. [7] based on acoustic emission signal proposed a modified time-dependent excitation (TDE) model to detect defects of angular contact ball bearings. Chen et al. [8] made contributions on low-speed rolling bearing fault detection with subspace embedded feature distribution alignment and Structural Risk Minimization framework based on acoustic emission signal. Maroua et al. [9] analyzed the performance of different kinds of rolling bearings under five fully formulated axle gear oils with different viscosity and different formulations. 

Those methods can extract incipient fault features from the original signal, which work as the input feature vector of classifier or as the indicator of rolling bearing incipient faults. However, the de-noise method has the disadvantage of weakening the fault information. In addition, these time-frequency domain methods cannot adaptively extract features, which lead to the weak ability of bearings early detection. 

In recent years, machine learning-based methods are widely applied in many industries. In reference [10], an impact time-frequency dictionary was built to extract signal features with short-time matching method first, and then support vector machine (SVM) worked as classifier for incipient fault states. The supervised local Fisher discriminant and K-nearest neighbor method were introduced for weak fault diagnosing in [11], in which they are working for feature reduction and incipient fault state classification respectively. Ocak et al. [12] proposed a Hidden Markov Model (HMM) for rolling bearing fault detection and diagnosis, which can identify and detect early failures by tracking the probability changes of the pre-trained HMM under normal conditions. These methods usually use part of the normal state data in the initial stage to establish a single classification model, or employ existing normal state samples to construct abnormal discrimination criteria. However, the bearing has certain noise in the normal state, this kind of methods cannot automatically adapt to the irregular data fluctuations caused by the various noise, which may cause false alarms.

In the past decade, deep learning has already become an efficient way to detect and diagnose fault in many fields [13,14,15,16,17]. According to the authors’ literature research, deep learning is still in its infancy on incipient fault detection. Lu et al. [18] used deep neural networks (DNN) and long short-term memory (LSTM) to construct an online data distribution estimator, and used the prediction bias value generated by the estimator to identify incipient fault location. A two-way Gated Recurrent Unit (GRU) network with local features is proposed for different types of faults to realize effective identification of incipient faults [19]. A new framework for rotor-bearing system fault diagnosis under varying working conditions is proposed in [20], it introduced stochastic pooling and Leaky rectified linear unit to overcome the training problems in classical CNN. Chen and et al. studied mixed faults diagnosis from multiple components by combining two 1-D convolutional neural networks (CNNs) [21]. Mao et al. [22] proposed an on-line detection method based on self-adaptive deep feature matching for incipient faults of rolling bearings. However, due to weak incipient fault, the ability of feature representation is poor. At the same time, due to the constraints of online application scenarios, the amount of available target-bearing data are limited. These methods have insufficient normal state data information, thus there are some certain obstacles in accurate decision boundary construction of online detection models.

Machine learning is used to dig out regular information from training data and learn pattern recognition knowledge. The parameter number of current deep neural networks is always very huge. It requires sufficient training data for training in order to obtain ideal results. In the case of a limited amount data, data enhancement technology can be used to increase the diversity of training data. Meanwhile, transformation operations can improve the feature representation ability of training data, therefore the feature information of training data is more sufficient and the problem of model over fitting can be avoided. Existing methods of data enhancement include: geometric transformation [23], color space enhancement [24], kernel filter [25], as well as generative adversarial networks [26] based on the idea of antagonistic thoughts and neural style transfer [27] and other methods. A geometric transformation is applied in [28] to solve the data imbalance problem for bearing fault detection. 

To solve these problems, an incipient fault detection method based on multi-scale Deep-SVDD model with data enhancement is proposed in this research. First, the training data are transformed into multiple subspaces through data enhancement technology. Second, the prototype clustering method is introduced to improve the robustness of features under the framework of regularized dual averaging (RDA) algorithm, and then a robust multi-scale Deep-SVDD hyper sphere model is constructed. Finally, the product of the probability that the transformed sample is located in its respective subspace is calculated as the anomaly score to achieve early online fault detection. The effectiveness of the proposed method is verified by experiments on the IEEE PHM Challenge 2012 bearing dataset and XJTU-SY dataset. The contribution of this paper can be summarized as follows. (1) A robust multi-scale Deep-SVDD hyper sphere model is proposed for online anomaly detection. The data information based on data enhancement technology is enriched. By extracting robust low-rank deep features, this method can enhance the capacity of multi-scale features representation and has good robustness. (2) An anomaly alarm indicator is built for online scenarios. This indicator is based on the robust low-rank features extraction, and then can measure abnormality. Therefore, this indicator is very effectively suitable for online applications. The details of this work are as follows.

## 2. Deep Support Vector Data Description

Deep support vector data description (Deep-SVDD) [29] is a representative method of using deep learning for anomaly detection in recent years. The nonlinear high-dimensional mapping is replaced by a neural network in this method, which improved the ability in dealing with high-dimensional and very large data sets. Deep-SVDD can take advantage of deep learning to deal with high-dimensional representation and processing of massive data. 

Deep-SVDD constructs a neural network mapping. The method minimizes the volume of the hyper sphere containing the data features in the network when solving, and obtains the high-dimensional space common feature representation of normal data. The objective function is:(1)$$\underset{W}{\min}\frac{1}{n}\sum_{i=1}^{n}{\left\|\phi (x_{i};W)-c\right\|}_{2}^{2}+\frac{\lambda }{2}\sum_{l=1}^{L}{\left\|w^{l}\right\|}_{F}^{2}$$

The objective function consists of two items. The first item is the quadratic loss of the distance between the penalty sample feature and the center of the hyper sphere, and the second is the regular item that constrains the network weight to prevent over fitting, where ϕ is neural network mapping function, $x_{i},i=1,2,\ldots ,n$ is the sample data, W is the set of weight parameters of the network, $W=\{w^{1},\ldots ,w^{L}\}$. c is the center of hyper sphere, L is the number of layers of the neural network, l={1,2,…,L}. λ is the hyper parameters that control the weight decay. $W^{l}$ is the weight of lth hidden layer.

Optimizing the first item lets the network learn parameters W such that data points are closely mapped to the center c, and optimizing the second item is to minimize the volume of the hypersphere. 

Center c is fixed in the neighborhood of the initial network outputs, which makes stochastic gradient descent (SGD) convergence faster and more robust [30]. 

The abnormal score of the Deep-SVDD algorithm evaluation sample can be calculated by the following:(2)$$s\left(x\right)={\left\|\left\|\phi (x_{i};W^{*})-c\right\|\right\|}^{2}$$
where $W^{*}$ is the network parameter of a trained model. 

The larger s(x), the farther the sample is from the center of the hyper sphere, the higher the degree of abnormality of the sample.

## 3. The Robust Multi-Scale Deep-SVDD Model of Incipient Fault Detection

In this section, the proposed incipient Fault detection method for bearing is divided into offline stage and online stage. In the offline stage, employing data enhancement technology to transform a small number of normal samples into multiple feature spaces, based on this, the prototype clustering loss and multi-hyper sphere Deep-SVDD center loss are introduced to train the robust multi-scale Deep-SVDD model, and obtain each transformed model in the feature space, the distance-based cross entropy is used to determine the distance score threshold of the normal period data. In the online stage, the test samples to be detected are subjected to the same transformation enhancement, and then they are put into the trained deep model to extract the deep features. The extracted deep features are used with each prototype center to calculate the distance-based anomaly score, and finally combined with the threshold value. When the score is less than the threshold, the sample is regarded as normal, otherwise, it is judged to be abnormal. Each step of the proposed robust multi-scale deep-SVDD model is elaborated in the following. The detailed flow chart is shown in Figure 1.

### 3.1. Signal Enhancement

Vibration signal is a special one-dimensional datum and there is no specific neighborhood or order. Thus, traditional geometric transformations such as translation and rotation cannot be performed. In order to enable the transformation-based method to process vibration signal data, we propose a data transformation method for vibration signals. Specifically, we propose two transformations of vibration signals from the perspective of graphics.

#### 3.1.1. Horizontal Scaling

First, we crop the length of p%(0≤p≤50) from either end of the original signal. To ensure the same dimension of the feature space after transformation, we use the resampling method to sample the cropped signal to the length of the original signal, which is equivalent to stretching the original signal in the horizontal direction from a graphical point of view. In addition, for a vibration signal sample, to reduce information loss, we cut the two ends of the signal to obtain two sets of data of equal length, which are used as the two channels of the transformed data, for signal samples are displayed in the same feature space at different scales. As shown in Figure 2, the original signal length is 1280, the cropping parameter p is set to 30, and the two channels of the transformed sample are obtained by cropping from the left and right ends respectively.

#### 3.1.2. Vertical Scaling

We set scaling parameters 0<α<1, transforming the original signal as follows:(3)$$f(x_{i})=\{\begin{array}{c} \left(1+\alpha \right)x_{i},x_{i}<0\\ \left(1-\alpha \right)x_{i},x_{i}\geq 0 \end{array}$$

Vertical scaling does not change the length of the signal and the signal can extend or shorten in the vertical direction from the graph. This transformation is also the display of different scales of the signal. Similarly, to ensure the consistency of the feature space after transformation, the samples after vertical transformation are set to be two channels.

By setting different parameters and different combinations of horizontal and vertical transformation, we can obtain a variety of transformation methods to process the original signal. In this paper, the original vibration signal sample space X is transformed to obtain M subspaces $X_{1},\ldots ,X_{M}$. The transformed sample is represented as T(x,i),…,T(x,M).

### 3.2. Prototype Clustering

Prototype clustering is a clustering algorithm that uses the prototype to represent the center of the cluster. The prototype clustering algorithm usually needs to initialize the prototype cluster center and then employs the idea of iterative solution to find the cluster prototype.

Learning Vector Quantization (LVQ) [31] is a typical prototype clustering method. The LVQ algorithm uses prototype vectors to represent clusters. The sample is assumed to be labeled, and then the label information is working as an aid in the iterative optimization process to find the optimal prototype vector, which represents the cluster structure. The high-dimensional clustering space is divided into n clusters, and each prototype vector represents a cluster. The solution steps of the LVQ algorithm are as follows (Algorithm 1):
**Algorithm 1** Learning Vector Quantization**Input:**Sample set $M=\left(x_{1},y_{1}\right),\left(x_{2},y_{2}\right),\ldots ,\left(x_{m},y_{m}\right)$; Suppose that, q is the number of prototype vectors, $t_{1},t_{2},\ldots ,t_{q}$ is the initial category of each prototype vector, and η∈(0,1) is the learning rate.**Ouput:**prototype vector $v_{1},v_{2},\ldots ,v_{q}$
1:Initialize the prototype vector $v_{1},v_{2},\ldots ,v_{q}$.2:**Loop**3: Select samples $\left(x_{j},y_{j}\right)$ from sample set M randomly.4: Calculate the distance between $x_{j}$ and $v_{i}(1\leq i\leq q)$: $d_{ji}={\left\|x_{i}-v_{i}\right\|}_{2}$
5: Find the prototype vector $v_{i}$ closest to $x_{j}$, $i^{*}=\arg\min_{i\in \{1,2,\ldots ,q\}}d_{ji}$
6: **If** $y_{i}=t_{i}$7:   
$v^{\prime }=v_{i^{*}}+\eta \left(x_{j}-v_{i^{*}}\right)$
8:
 **Else**
9:   
$v^{\prime }=v_{i^{*}}-\eta \left(x_{j}-v_{i^{*}}\right)$
10:
 **End if**
11:   Update $v_{i}$ to $v^{\prime }$
12:**Until** the stop criterian is reached

The algorithm finally learns a set of prototype vectors. Moreover, each of them represents the center of a certain area, which is equivalent to the center point of Voronoi division in space geometry. This center point is the center point of transformed sample in the neural network feature space.

### 3.3. Distance-Based Cross Entropy Loss

The original sample space X undergoes M transformations to obtain the transformed sample T(x,i),…,T(x,M). For each transformed sample T(x,i), calculate the following conditional probability:(4)$$p(T(x,i)\in X_{i})=\frac{e^{-{\left\|E_{\theta }(T(x,i))-c_{i}\right\|}_{2}^{2}}}{\sum_{i=1}^{M}e^{-{\left\|E_{\theta }(T(x,i))-c_{i}\right\|}_{2}^{2}}}$$
where $E_{\theta }$ is the network for feature extraction, $c_{i}$ is the center of $X_{i}$.

The distance-based cross entropy (dce) is expressed as:(5)$$loss_{dce}=-\log p(T(x,i)\in X_{i})$$

Minimizing distance-based cross-entropy loss can map data samples to the class feature space near the prototype center, and improve the separability between classes. Compared with the softmax loss in traditional neural networks, it is more robust.

### 3.4. Robust Multi-Scale Deep-SVDD

The main idea of the robust multi-scale Deep-SVDD method proposed in this section is to perform data enhancement on the normal samples in the single-class anomaly detection and generate multiple transformations to construct SVDD hyper spheres, and use each transformation in multiple Deep-SVDD hyper spheres. The comprehensive score is used to measure the degree of sample abnormality.

First, to improve the robustness of feature extraction, we select robust deep auto-encoding as the main framework, in which the robust deep auto-encoding encoder is conducted for feature extraction, so as to map the original samples to the low-rank feature space. Second, the learning vector method is used to find out the prototype centers $c_{1},\ldots ,c_{M}$ of the transformed M samples subspace in the robust deep auto-encoder low-rank space. On this basis, the Deep-SVDD center loss is added, so that all normal samples are as close as possible to the center of each prototype, and the intra-class aggregation degree of each subspace is constrained. The final optimization function is as follows:(6)$$\begin{array}{l} \underset{\theta ,S}{\min}{\left\|L_{D}-D_{\theta }\left(E_{\theta }\left(L_{D}\right)\right)\right\|}_{2}^{2}+loss_{dce}+\mu {\left\|E_{\theta }\left(L_{D}\right)-c_{i}\right\|}_{2}^{2}+\lambda {\left\|S\right\|}_{2,1}\\ s.t.\;\;X-L_{D}-S=0 \end{array}$$
where μ>0, λ>0 are regularization coefficient. Increasing the value of μ will make the normal sample features move closer to the center of each prototype, and vice versa, it will weaken the effect of the features gathering to the center.

### 3.5. Calculation of Anomaly Score

After the above steps, the training model can extract features from the input sample data after specific transformations and obtain the corresponding set of prototype centers $c_{1},\ldots ,c_{M}$, and then we can measure the degree of abnormality of the test sample. In the test stage, the test sample x undergoes M transformations to obtain the transformed sample T(x,i),…,T(x,M). Put the transformed samples into a robust deep self-encoding encoder to extract features, according to Formula (4), the distance between all transformed samples and the centers of all prototypes is calculated to obtain the probability that they are located in their respective subspaces. Then the probability of the test normal sample x is the product of the probabilities that all the transformed samples are located in their respective subspaces, and the final anomaly score is expressed as:(7)$$\mathrm{Score}\left(x\right)=-\sum_{i}^{M}\log P\left(T(x,i)\in X_{i}\right)$$
where the score represents the abnormal score of test sample x. The higher the score, the more abnormal.

Finally, for the bearing incipient fault detection, we need to determine a threshold for the abnormal score of a normal sample to determine whether the calculated abnormal score of the test sample meets the abnormal standard, that is, whether the bearing operating state is abnormal. In this paper, the maximum value of the training data anomaly score is directly used as the threshold standard. 

## 4. Experiment

Experiments on the IEEE PHM Challenge and XJTU-SY datasets are performed to verify the effectiveness of the proposed method. The programming environment is Python 3.6.0, Guido van Rossum, Beijing, China. The computer used in the experiment is configured with i5-8400 processor and 16 G memory.

### 4.1. Dataset Introduction

IEEE PHM Challenge 2012 dataset is collected from the PRONOSTIA platform (shown in Figure 3a) [32], which specially designed and implemented by the AS2M department of the French FEMTO-ST Institute. It provides the entire life cycle data of rolling bearings through accelerated life degradation experiments. Bearings are working under three different working conditions in these experiments, (1) the engine speed is 1800 rpm and the load is 4000 N, (2) the engine speed is 1650 rpm and the load is 4200 N, (3) the engine speed is 1500 rpm and the load is 5000 N.

The XJTU-SY dataset is provided by the Institute of Design Science and Basic Component at Xi’an Jiaotong University (XJTU) [33], China and the Changxing Sumyoung Technology Co., Ltd. (SY), Zhejiang province, China. The platform is shown in Figure 3b. Three kinds of experimental working conditions were designed in this experiment, and five bearings were tested in each working condition. (1) The engine speed is 2100 rpm and the load is 12 kN. (2) The engine speed is 2250 rpm and the load is 11 kN. (3) The speed is 2400 rpm and the load is 10 kN. 

### 4.2. Model Parameter Settings

The same data transformation (Horizontal and Vertical scaling) is conducted in the experiments. The parameter of Horizontal scale p is set to be 16, the value set is {0, 2, 4, 6, 8, 11, 14, 17, 20, 23, 27, 31, 35, 39, 43, 47}. The parameter of vertical scale α is set to be 3, and the value set is {0, 0.3, 0.7}. There are 48 combinations of these two transformations. The neural network structure used in the experiment is a deep residual network [34]. In the multi-scale robust Deep-SVDD, μ =0.0001,λ =0.5, γ=0.002, the training iteration number is 100, and the size of each training batch is 8.

### 4.3. Incipient Fault Detection Results

Bearing 1_2 and bearing 1_3 in the IEEE PHM Challenge 2012 dataset, as well as the bearing 1_1 and the bearing 2_2 in the XJTU-SY dataset, are the target bearings, as shown in Table 1. The first 100 samples are selected for data transformation to obtain signals of 48 different scales in this experiment, and then the obtained data are put into the multi-scale robust Deep-SVDD model for training to complete the model training. In the test stage, the test samples are first subjected to data transformation, and then input into the model to calculate the abnormal score of each sample. The results of abnormal score and the RMS value are shown in Figure 4 and Figure 5.

### 4.4. Comparative Results

To verify the superiority of the proposed algorithm, comparison between five other widely used methods for incipient fault diagnosis and detection and the proposed method is made. Among them, bandwidth empirical mode decomposition and adaptive multiscale morphological analysis (BEMD-AMMA) [35] is a typical method based on weak signal analysis, local outlier factor (LOF) and isolation forest (iFOREST) are two classic anomaly detection algorithms, meanwhile, Self-Adaptive Deep Feature Matching method (SDFM) [22] and Sparse Dictionary Representation (SDR) [36] methods are also used for comparison. 

The Spectrum of bearing fault at different sample points are shown in Figure 6, where Figure 6a is for PHM1_3, and Figure 6a is for XJTU1_1. As we can see from both Figure 6a,b, the fault frequency gradually shows up with time. 

We define a deviation rate of incipient fault detection (DA) to evaluate the methods’ performance mentioned above.
(8)$$DA=\frac{\left|p_{d}-p_{r}\right|}{p_{e}-p_{r}}\times 100\%$$
where is the detected sample point of incipient fault through method, $p_{r}$ is the real sample point of incipient fault, and $p_{e}$ is the end sample point of bearing in whole life. 

The anomaly detection result is shown in Table 2.

As shown in Table 2, the detection result of proposed method is the best one in the comparison. It indicates that employing multi-scale signal samples can enhance feature representation and make incipient fault more sensitive. The robust low-rank deep features extracted by multi-scale robust Deep-SVDD hyper sphere model have strong anti-noise ability for signal fluctuations. Thus, the stability and accuracy of detection results are relatively high.

## 5. Conclusions

This paper proposes a multi-scale robust incipient fault detection method of rolling bearing with data enhancement. The data enhancement technology is incorporated into the framework of the robust deep auto-encoding network, to improve the anti-noise ability. It makes the extracted features more robust. Moreover, the constructed robust multi-scale Deep-SVDD model is with good stability by adopting the multi-scale vibration signal features. From the experimental results, the proposed method is more sensitive to incipient faults and has lower false alarm number. The proposed method significantly improves the performance of incipient fault detection of rolling bearings.

## Figures and Tables

**Figure 1 sensors-22-05681-f001:**
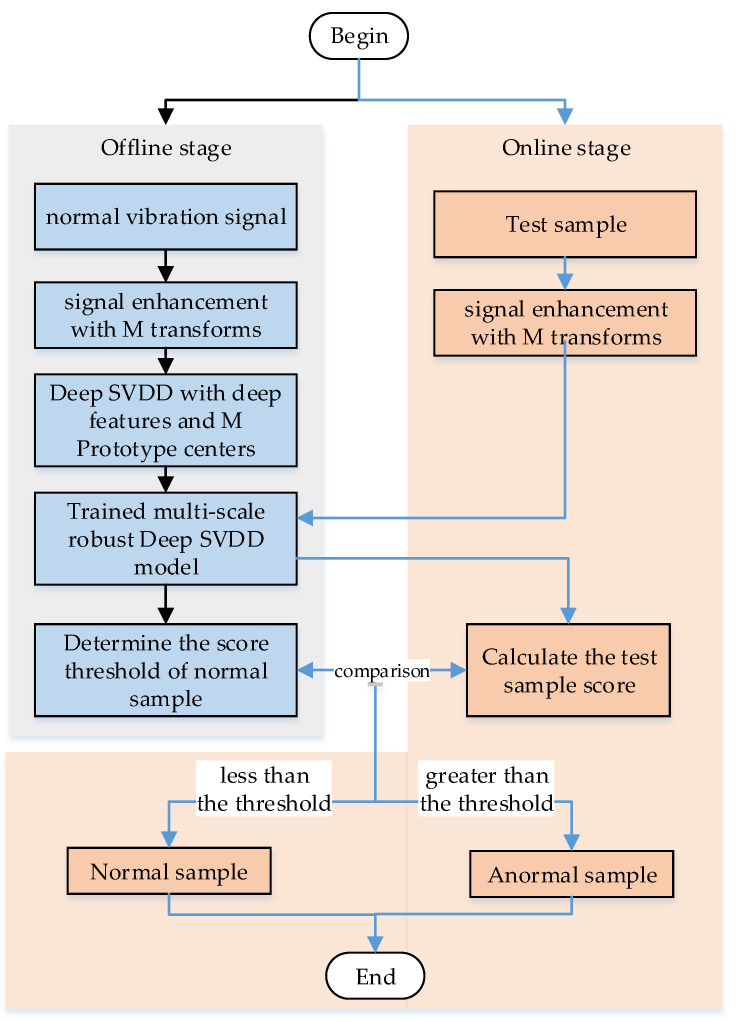
Flowchart of the Robust Multi-scale Deep-SVDD Model.

**Figure 2 sensors-22-05681-f002:**
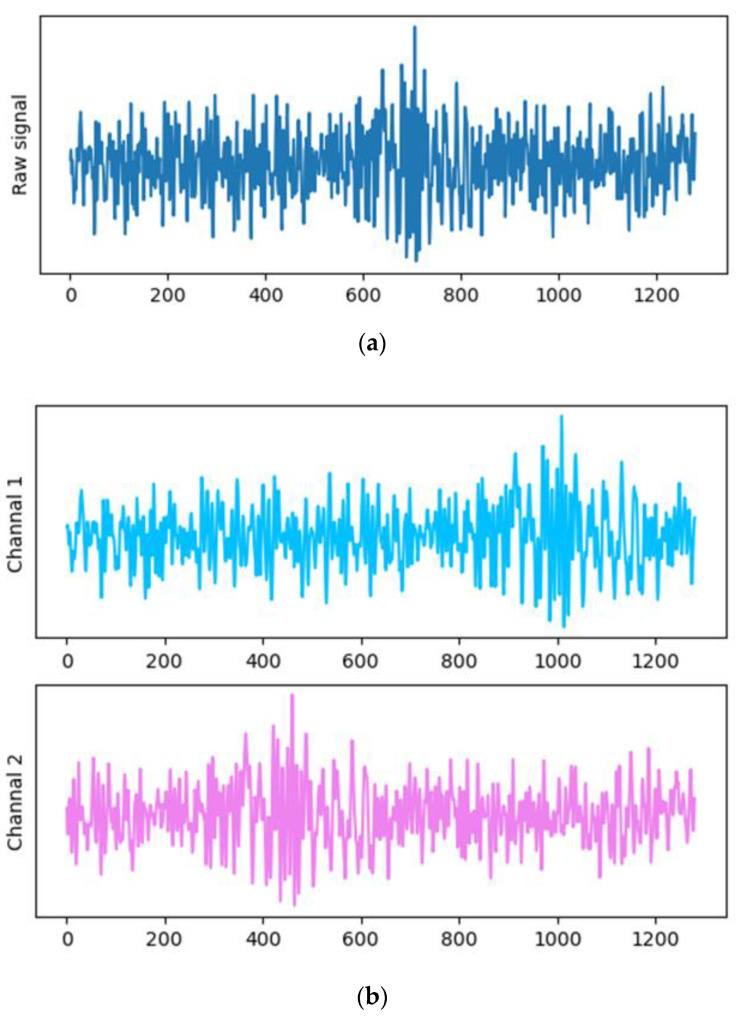
Schematic diagram of the horizontal scaling of the vibration signal with (**a**) the original signal and (**b**) two transformed channel signals.

**Figure 3 sensors-22-05681-f003:**
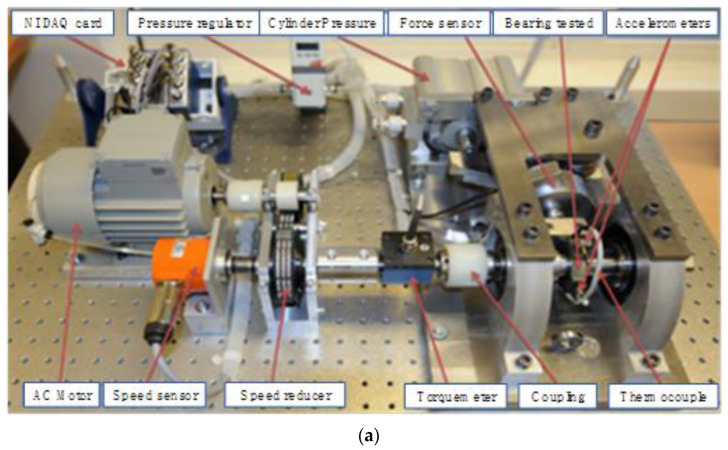
Test platforms of (**a**) PRONOSTIA [32] and (**b**) XJTU-SY [33].

**Figure 4 sensors-22-05681-f004:**
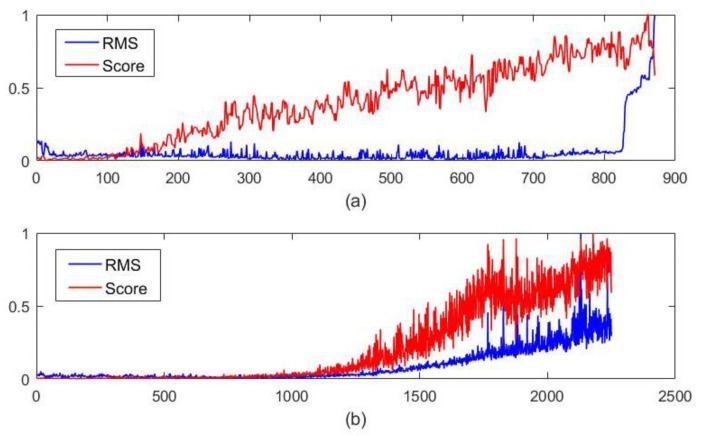
Result comparison of abnormal score and the RMS value with (**a**) PHM1_2 and (**b**) PHM1_3.

**Figure 5 sensors-22-05681-f005:**
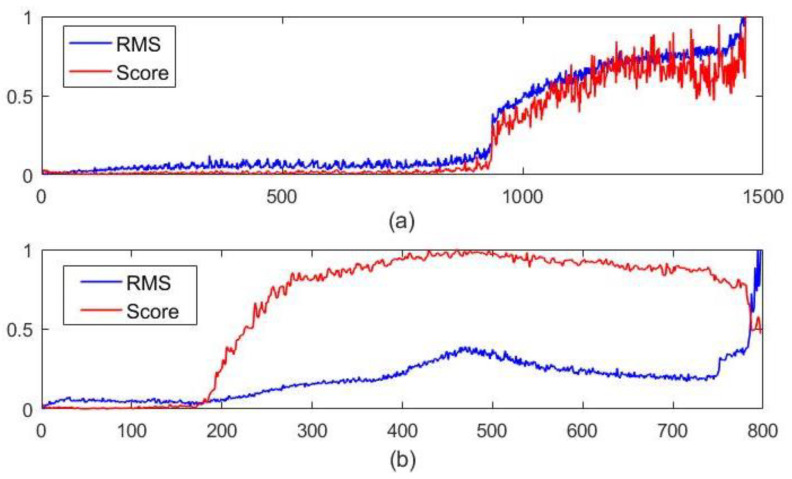
The comparison results of abnormal score and the RMS value with (**a**) XJTU1_1 and (**b**) XJTU 2_2.

**Figure 6 sensors-22-05681-f006:**
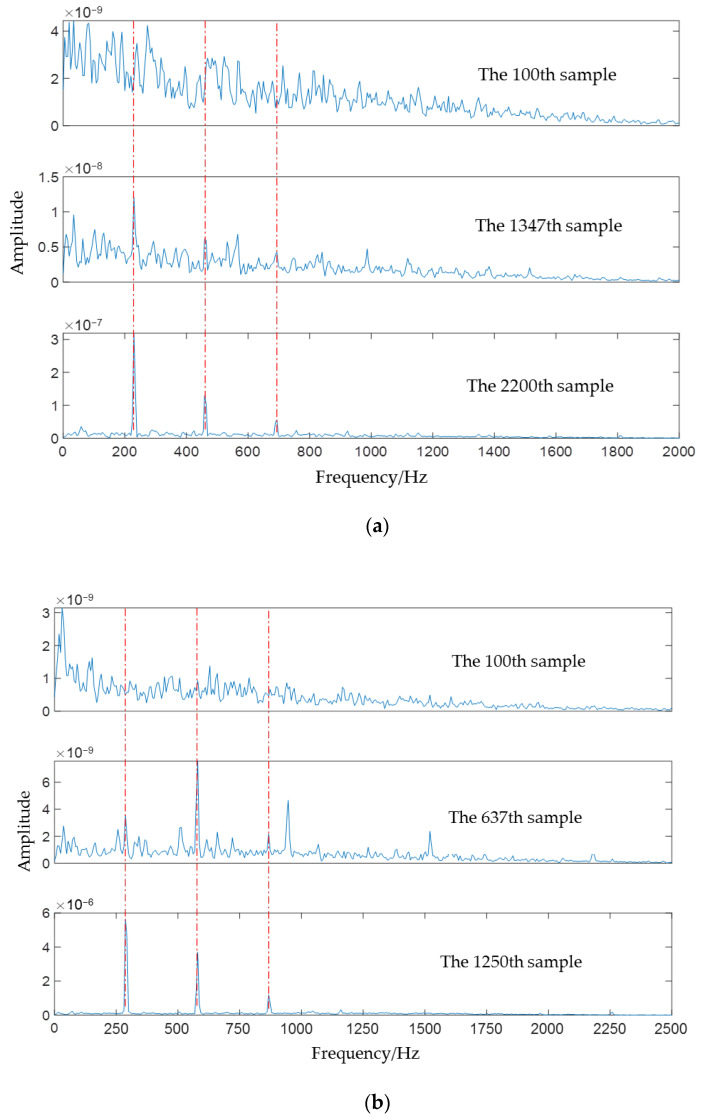
Spectrum of bearing fault at different sample points (**a**) PHM1_3 (**b**) XJTU1_1.

**Table 1 sensors-22-05681-t001:** Experiment dataset.

Dataset	Sample	Number of Sample	Training Sample	Testing Sample	The Real Sample Point of Incipient Fault	Number of Early Fault Samples
IEEE PHM Challenge 2012 dataset	Condition 1 Bearing1_2	871	The first 100 samples	The rest 771 samples	-	479
	Condition 1 Bearing1_3	2375	The first 100 samples	The rest 2275 samples	1348th	1027
XJTU-SY dataset	Condition 1 Bearing1_1	1476	The first 100 samples	The rest 1376 samples	634th	839
	Condition 2 Bearing2_2	1932	The first 100 samples	The rest 1832 samples	-	942

**Table 2 sensors-22-05681-t002:** Comparison of anomaly detection results.

Comparison Methods	PHM1_3		XJTU1_1	
	The Detected Sample Point	Deviation Rate of Incipient Fault Detection	The Detected Sample Point	Deviation Rate of Incipient Fault Detection
1. BEMD-AMMA	1600	55.79%	1320	57.33%
2. LOF	1236	20.35%	944	12.51%
3. iFOREST	1341	30.57%	1041	24.08%
4. SDFM	1156	12.56%	1137	35.52%
5. SRD	1160	12.95%	1013	20.74%
6. The proposed method	997	2.9%	826	1.55%

## Data Availability

Not applicable.

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
