# Peer review of "The Robust Multi-Scale Deep-SVDD Model for Anomaly Online Detection of Rolling Bearings"

_sensors, 2022, doi:10.3390/s22155681_

Round 1

Reviewer 1 Report

This manuscript presented a deep learning model for rolling bearings fault detection. The manuscript has its merits but there are a few issues that require major revision before further consideration.

1. The introduction mainly introduces the literature regarding using traditional machine learning approaches and DNNs to solve the problem of fault diagnosis. However, the discussion over related works about the main contribution, data enhancement and knowledge transfer, of this paper is missing. Also, more literature reviews on the paper from top journals (e.g. TII and TMECH) and be discussed.

2. The equations in the paper are very unclear, many variables are listed without explanation. For example, x_i and W^l in equation 1; L, D, E in equation 5.

3. Please explain why Equation 1 is consist of two items?

4. How does c in Equation 1 calculated.

5. Both equation 2 and equation 7 are described as "abnormal score", what is the difference?

6. How does equations 4 and 5 connected?

7. The detailed mode framework is unclear, a detailed algorithm or flowchart will help to improve the readability.

8. Ablation studies are needed to validate the effectiveness of each component of the loss functions.

Reviewer 2 Report

Data enhancement achieved through transfer into multiple subspaces; prototype clustering method used to enhance the robustness of features representation; m.s. Deep-SVDD hypersphere model constructed for the online detection of anomalies of rolling bearings. Experiments performed on two data sets. 

The paper is compact, well-written, and the results look sound.

I recommend the citation of Fu L. et al., Appl. Sci. 12, 5240 (2022) Bearing Cog... https://doi.org/10.3390/app12105240 

Reviewer 3 Report

I found your article very interesting, but in my opinion below remarks would improve your manuscript under the scientific level.

Comments and Suggestions for Authors:

11. In the Introduction I don’t agree with the sentence: “There are mainly two kinds of incipient fault detection methods, signal analysis-based methods and machine learning-based methods.” Still, the ML model must be based on the data obtained from the dynamical response, so that is why I controvert with above mentioned statement.

22. I suggest to extend the list of references with papers referring to the detection of other operational parameters in rolling-element bearings, which can change in time:

·       Liu et al. (2021) – A combined acoustic and dynamic model of a defective ball bearing – Journal of Sound and Vibration 501, 116029.

·       Ambrożkiewicz et al. (2022) – The influence of the radial internal clearance on the dynamic response of self-aligning ball bearings – Mechanical Systems and Signal Processing 171(8), 108954.

·       Hammami et al. (2018) – Friction torque in rolling bearings lubricated with axle gear oils – Tribology International 119, 419-435.

33. It is worth adding the Remainder in the end of Introduction section.

44. In the manuscript I don’t see very important information on the number of samples, how they are distinguished and what is the model preparation, i.e. what amount of data is taken for the training and validation.

55. I suggest to put the experiment data into the table.

66. In Figure 3a, captions are not visible.

77. I’m not sure, but information in Table 1, should be put in the form of confusion matrix.

I 8. I don’t see really important issue, that how do you define fault of bearing? There is no reference to the characteristic frequencies, which easily detect the fault.

99. From the manuscript I don’t see the real advantage of your approach in comparison with other ML algorithms. I suggest to compare it directly with other mentioned algorithms.

110.   Why for programming do you use Matlab and Python in the same time? I don’t see the point.

Round 2

Reviewer 1 Report

The authors have addressed my previous comments I have no further questions.

Reviewer 3 Report

Dear Authors,

I see, that all remarks have been introduced increasing the scientific quality of the paper. By introducing the Remainder, I meant it in a little bit different form, but it doesn't diminish the content of the research. In the present form I can recommend it for publishing.

Yours faithfully,

Reviewer

This manuscript is a resubmission of an earlier submission. The following is a list of the peer review reports and author responses from that submission.